# Schrödinger’s Cat Meets Occam’s Razor

**DOI:** 10.3390/e24111586

**Published:** 2022-11-01

**Authors:** Richard D. Gill

**Affiliations:** Mathematical Institute, Leiden University, P.O. Box 9512, 2300 RA Leiden, The Netherlands; gill@math.leidenuniv.nl

**Keywords:** the measurement problem, Schrödinger’s cat, C*-algebras, von Neumann algebras

## Abstract

We discuss V.P. Belavkin’s approach to the Schrödinger cat problem and show its close relation to ideas based on superselection and interaction with the environment developed by N.P. Landsman. The purpose of the paper is to explain these ideas in the most simple possible context, namely: discrete time and separable Hilbert spaces, in order to make them accessible to those coming from the philosophy of science and not too happy with idiosyncratic notation and terminology and sophisticated mathematical tools. Conventional elementary mathematical descriptions of quantum mechanics take “measurement” to be a primitive concept. Paradoxes arise when we choose to consider smaller or larger systems as measurement devices in their own right, by making different and apparently arbitrary choices of location of the “Heisenberg cut”. Various quantum interpretations have different resolutions of the paradox. In Belavkin’s approach, the classical world around us does really exist, and it evolves stochastically and dynamically in time according to probability laws following from successive applications of the Born law. It is a collapse theory. The quantum/classical distinction is determined by the arrow of time. The underlying unitary evolution of the wave-function of the universe enables the designation of a collection of beables which grows as time evolves, and which therefore can be assigned random, classical trajectories. In a slogan: the past is particles, the future is a wave. We, living in the now, are located on the cutting edge between past and future.

## 1. Introduction, and Parental Advisory

Some of those who consider themselves true physicists may find what I have to say in this paper at best meaningless or stupid, at worst heretical. I am happy with the notions that quantum mechanics is non-local, that the physics of quantum mechanics is not time-reversible, and that it involves irreducible randomness. It seems to me (a mathematician and an applied statistician) that that is the message which quantum mechanics has been shouting at us since its birth: namely, that that is what reality truly is like. Obviously, this contradicts what by the end of the 19th century was dogmatic. The wonderful thing of the Växjö conferences has been that people interested in quantum foundations have been able to say such terrible things to one another and remain friends.

I want to introduce my take on ideas of Belavkin (2007) [1] and Landsman (1995) [2] which I find very closely related and very attractive. Belavkin is unfortunately no longer with us; Landsman by now has quite different ideas; see their book Landsman (2017) [3].

The paper is about the Schrödinger cat paradox, which, I believe, is indeed truly only a paradox, since (as a purely mathematical issue) it can be easily resolved. This does have consequences for physics. We have a mathematical framework which might be applied to a physical context in different ways. How to apply it in a given context is a question for physicists to decide. Converting a bug into a feature, if there is some choice as to how a certain framework can be applied to a real world physical problem, one can also say that this offers a challenge to experimenters to find out whether experimental results can be used to restrict that choice. In our case, if wave function collapse is real and non-local and due to gravity, experiment should be able to tell us more about it.

It is very important to distinguish between the mathematical models which mathematical physicists develop, and the real world itself. The models are supposed to describe the real world, and aid us in engineering it. Such a model helps one to *understand* the phenomenon being modelled, in the sense that it makes one comfortable with it and aids in dealing with it, but it is clear that quantum mechanics contains elements which seem to conflict with very basic human understanding. We evolved in a real world with a lot of randomness, but we find randomness horrific, since our evolutionary speciality is to predict what might happen next and base our actions on choices between different possible futures. We associate randomness with the choices of the Gods, and they may not be benevolent.

Both Belavkin and Landsman argued for including the mathematical existence of a classical-like world as part of the axiomatic foundations of quantum mechanics. For them, classicality is not merely an emergent phenomenon. Landsman (1995) went on to distinguish two kinds of classicality, or attitudes to it. He wrote “those believing that the classical world exists intrinsically and absolutely (such persons later termed by them B-realists) are advised against reading this paper”. He adopts a milder position, calling it that of an A-realist: we live in a classical world but to give it special status is like insisting that the Earth is the centre of the universe. The B-realists are accused of living under some kind of hallucination. Landsman presents arguments pointing in a particular direction to a resolution of the measurement problem which at least would satisfy the A-realists. We point out in this paper that the theory earlier developed by Belavkin (surveyed in their 2007 paper) seems to complete Landsman’s program or at least exhibits a “realisation” satisfying their desiderata. At the same time it seems that this completion of the program ends up giving both A- and B-realists equal licence to accuse the others of living under hallucinations. I think that this distinction is a philosophical distinction and as much an aesthetic issue as anything else.

The theory is presented in the context of a standard (separable) Hilbert space description of quantum theory. The author is well aware that to go further one will sooner or later have to leave Hilbert spaces behind one, and enter a more abstract and exotic mathematical universe, to do justice to the nature of our real universe.

## 2. The Basic Framework

Belavkin’s eventum mechanics, developed in the 80’s, and a recent exposition of which is given by Belavkin (2007), has been created in an attempt to resolve the Schrödinger cat problem by showing that measurement, and random collapse of the wave function, can be seen as the result of a deterministic unitary evolution as long as one recognises that this evolution must take place in a mixed classical-quantum system. The collapse of the wave function is the stochastic result of a deterministic, unitary evolution in a situation where there is a quantum interaction between a quantum system and a classical system. Classicality corresponds to a superselection rule, saying that not all observables (in the sense of bounded operators) are actually observable (in the sense that quantum superposition of certain states cannot occur, or at least, can never be detected). The essential and unorthodox aspect of the theory is that it is time irreversible. Unitarity is retained but in the Heisenberg picture, the time evolution of the relevant observables is an endomorphism, not an isomorphism.

We present Belavkin’s basic theory in the most simple possible mathematical context, involving nothing more elaborate than separable Hilbert spaces K and L and their tensor product H=K⊗L. There will be a unitary operation *U* and an initial pure (vector) state |Ψ〉 defined on H. The initial state, together with iterated application of *U* on the initial state vector, define a discrete time evolution of the system. The evolution can also be traced backwards in discrete time, through iterates of $U^{*}$. To be honest, though, we do need to make use of some elementary properties of *von Neumann algebras*. Here is the definition. Given a collection A of bounded operators on H, its commutant $A^{\prime }$ is the collection of all bounded operators on H which commute with every element of A. A von Neumann algebra is a collection of bounded operators on the Hilbert space, whose bicommutant is equal to itself: $A^{\prime \prime }=A$. Concrete examples are given later.

We suppose that K has a particular orthonormal basis denoted by |x〉,x∈X, where the index set X is finite or countably infinite. Let |i〉,i∈I denote an arbitrary (finite or countably infinite) orthonormal basis of L so that the kets |x,i〉 form an o.n.b. of H. The coordinate x∈X is supposed to indicate the (classical) state of the real world, which evolves stochastically by its interaction with an underlying quantum world. In order for this to be meaningful we must make some assumptions relating *U* to the product structure of H and to the preferred basis of K.

Later we will discuss the more general situation in which H is not necessarily (a priori) of a product form, and in which a “preferred basis” of part of this space emerges, though in general not uniquely, from the other physical information about the system: specification of *U*.

We next introduce certain algebras of bounded operators on K, L and their product H=K⊗L. A *-algebra of bounded operators of a Hilbert space H is a subset of B(H), closed under addition, multiplication (composition of operators), scalar multiplication with complex numbers, and the involution * (adjoint). It is called a $C^{*}$-algebra when it is a closed subset of B(H) with respect to the operator norm; hence it is also complete for the topology induced by this norm. It is called a von Neumann algebra it is furthermore closed with respect to the weakest topology making the “matrix elements” 〈ϕ|Aψ〉 continuous for all ϕ, ψ. According to von Neumann’s bicommutant theorem, this is equivalent to $A^{\prime \prime }=A$. Abstract versions of all these objects also exist; in particular, the abstract version of a von Neumann algebra is called a $W^{*}$-algebra. The reason we must insist on von Neumann algebras is that normal states—states satisfying a natural continuity property—can be represented by density matrices: trace class operators on H. It has been said that $C^{*}$-algebras form the right context for non-commutative geometry; von Neumann algebras the context for non-commutative probability. Firstly, define
$$C_{K,X}\;=\;\left\{\sum_{x}c_{x}|x\rangle \langle x|:{\left(c_{x}\right)}_{x\in X}\;\mathrm{is}\;a\;\mathrm{bounded}\;\mathrm{sequence}\;\mathrm{of}\;\mathrm{complex}\;\mathrm{numbers}\right\}.$$
Using a prime to denote the commutant of a set of bounded operators, i.e., the set of bounded operators each of which commutes with everything in the first set, one can verify by direct calculation that $C_{K,X}^{\prime }=C_{K,X}$ and hence
$$C_{K,X}^{\prime \prime }=C_{K,X}^{\prime }=C_{K,X}.$$
It follows from von Neumann’s double commutant theorem that $C_{K,X}\subseteq B\left(K\right)$ is a von Neumann algebra: that is to say, a $C^{*}$-algebra which is closed under the weak norm topology. The elements of $C_{K,X}$ all commute with one another, and $C_{K,X}$ is maximal in the sense that no other element of B(K) commutes with all of $C_{K,X}$.

Now define
$$C\;=\;C_{K,X}⊗C1_{L}\;\subseteq \;B\left(H\right),$$
$$A=C_{K,X}⊗B\left(L\right)\;\subseteq \;B\left(H\right).$$

The tensor product of von Neumann algebras is the smallest von Neumann algebra containing the tensor products of individual elements of the two algebras. $C1_{L}$ is the von Neumann algebra of all complex multiples of the identity on L. One can verify by direct calculation (or by appeal to general theory of von Neumann algebras: the commutant of a tensor product is the tensor product of the commutants) that
$$A^{\prime }\;=\;C,\;C^{\prime }\;=\;A$$
Note that C is an algebra of commuting observables, and as such it is maximal. We also have
C⊆A⊆B(H).

We shall refer to the elements of B(H) as *observables*. The word observable is just used for convenience. We are setting up a toy universe in which there are no external observers or measurements. There are just places or sectors in this universe, which we will call “worlds”, which have rich enough properties that they support “life as we know it”. One can imagine physical objects called observers living in such a world (maybe even imagining that with free will they can choose to do various different measurements), who see a consistent stochastically time-evolving environment. Yet these observers and their measurement devices are subject to the same laws of quantum physics as everything else.

We think of C as being a set of *beables*, that is to say, physical quantities which can be given definite values. Very soon we will add a condition so that C becomes a set of *viables*, that is to say, the beables also have definite time trajectories, or lives. We see C as a possible (classical) world which can be found within the quantum universe given by *U* and H.

The commutant of C, the algebra A, is the corresponding set of *predictables*. As we will see, these observables have definite probability distributions relative to C, and moreover are used to predict the stochastic future of the beables.

Now consider a unitary operator *U* acting on H; so $U^{*}U=UU^{*}=1$. A pure state vector |Ψ〉∈H is mapped by *U* to U|Ψ〉. We will use *U* to define a discrete time dynamics on our system, corresponding to iterated application of *U*. We say that this time evolution is *compatible* with the classical-quantum pair C, A if
$$U^{*}AU\;\subseteq \;A,$$
which can be easily shown to be equivalent to
$$UCU^{*}\;\subseteq \;C.$$

The key to this equivalence is to note that since *U* maps products to products, it maps commuting variables to commuting variables. Note that in the Heisenberg picture, an observable *A* is mapped to $U^{*}AU$ by one *forwards* time-step, and an observable *C* is mapped to $UCU^{*}$ by one *backwards* time-step. Thus, the assumption just made states that future predicables are also predictable, and past beables are also beable. Rather neatly, one need only assume that future predictables are also predictable *or* that past beables are also beable. A classically-minded physicist would prefer the latter; a quantum-minded physicist the former. The other property comes for free.

Belavkin uses the word non-demolition property, or causality property, rather than compatibility. We here use the word compatibility because later we will consider H and *U* as given, the essential physics of our toy universe. The Hilbert space H will not a priori be given any special structure. Given H and *U*, a particular choice of von Neumann subalgebras C⊆A⊆B(H) having the properties $C^{\prime }=A$, $A^{\prime }=C$, if also compatible with *U*, then supports a viable quantum-classical world which is part of the universe. There can be many such worlds.

Incidentally, one can again start with assumptions for the classical-minded and assumptions for the quantum-minded. The classical-minded physicist starts with a commuting von Neumann algebra C, defines $A=C^{\prime }$; he automatically obtains C⊆A and the other relations between A and C. The quantum minded physicist starts with a von Neumann algebra A which is such that $A^{\prime }=C$ is commutative; she similarly obtains all the other relations between the two algebras.

A state on the product space H can be represented in the ordinary way with a (bipartite) density matrix ρ, and the expectation value of an arbitary observable *B* is trace(ρB). We want to restrict the state to the algebra A. We wish to think of the state abstractly as the mapping from a given sub-algebra (in particular, A) of observables to their expectation values. In that context, different density matrices can be indistinguishable from one another, i.e., generate the same expectation values.

Denote by [x] the subspace of all vectors of the product space of the form |x〉⊗|ψ〉, with |ψ〉∈L arbitrary, and $\Pi_{x}$ as the orthogonal projection onto this space, then with respect to the algebra A, a bipartite density matrix ρ cannot be distinguished from $\sum_{x}\Pi_{x}\rho \Pi_{x}$. Normalizing each component of this sum, the state $\Pi_{x}\rho \Pi_{x}/\mathrm{trace}\left(\rho \Pi_{x}\right)$ lives on the subspace [x] which is just a copy of L. Write $p_{x}=\mathrm{trace}(\rho \Pi_{x}$) and $\sigma_{x}$ as the just mentioned normalized state, thought of as a density matrix on L. Together, these remarks mean that any quantum state on A can be represented uniquely with the density matrix $\sum_{x}p_{x}\delta_{x}⊗\sigma_{x}$, where $\delta_{x}=\left|x\rangle \langle x\right|$, and the $p_{x}$ form a probability distribution on X. The states $\delta_{x}$ are pure states on $C_{0}$—they cannot be written as mixtures of other states (where states are taken to be “expectation values” defined on $C_{0}$).

[x] can also be called a *sector* of H and corresponds to a *superselection rule*: quantum superpositions between different eigenstates of a corresponding (possibly unbounded) observable *X* on H are impossible. In our product construction, a particular superselection rule was put into the model by hand. However, if we just consider H and *U* as given, different choices of C and A can be identified, compatible with the given H and *U*. Thus, different, mutually incompatible, superselection rules can be identified. However, if one would make some partial requirements on A or C, it is possible that their complete identity would then be fixed; in other words, a superselection rule can *emerge* from the physics. We need in advance to specify *time*, the unitary *U*, and we need to specify a weak kind of *locality* corresponding to a notion of *space*, in the form of some commutation properties. Principles of causality then determine what the classical world looks like.

Inspection of how an observable in A transforms under *U* reveals that *U* must have the following structure: written out blockwise with blocks $U_{xy}$, which are operators on L, indexed row-wise and column-wise by x,y∈X, for each *y* there must exist at least one *x* such that $U_{xy}\neq 0$; for each *x* there exists exactly one *y* with $U_{xy}\neq 0$. Thus, there exists a function *f* from X*onto* itself such that $U_{xy}\neq 0$ if and only iff f(x)=y. (For any x,i,j, the observable |x〉〈x|⊗|i〉〈j| must transform into a linear combination of observables of the same type.)

The unitarity of *U* implies that the $U_{xy}$ satisfy $\sum_{x:f(x)=y}U_{xy}^{*}U_{xy}=1$, $U_{xy}U_{xy}^{*}=1$ if f(x)=y, $U_{xy}U_{x^{\prime }y}^{*}=0$ if $f\left(x\right)=y=f\left(x^{\prime }\right)$, $x\neq x^{\prime }$. Conversely, given any *f* and $U_{xy}$ satisfying these properties, we can reconstruct *U*, compatible with A.

The (forwards) Heisenberg evolution of observables in A can be described through the mapping
$$\delta_{x}⊗B\;\;\;\mapsto \;\;\;\delta_{f(x)}⊗\;\;U_{x,f(x)}^{*}BU_{x,f(x)}.$$
The (forwards) Schrödinger evolution of states on A is similarly described through
(1)$$\delta_{y}⊗\sigma \;\;\;\mapsto \;\;\;\sum_{x:f(x)=y}\mathrm{trace}\left(\sigma U_{xy}^{*}U_{xy}\right)\;\delta_{y}⊗\;\frac{U_{xy}\sigma U_{xy}^{*}}{\mathrm{trace}(\sigma U_{xy}^{*}U_{xy})}.$$

The interpretation of this change of state is that the classical coordinate *y* jumps to one of the coordinates *x* with f(x)=y with probability $\mathrm{trace}(\sigma U_{xy}^{*}U_{xy})$ while the state on L is transformed into $U_{xy}\sigma U_{xy}^{*}$, normalized.

I view Equation (Equation 1), the most important result in this paper, as describing spontaneous collapse. In fact, the formula quite simply expresses iterations of the Lüders-von Neumann rule for the probabilities of outcomes and resulting transformation of quantum state, applied to successive observation of a sequence of observables of components of a composite quantum system. The probabilities are those given by the Born rule, the state is transformed by projection onto the eigenstate corresponding to which eigenvalue *x* has been realised (by the Lüders-von Neumann rule). Back to von Neumann (1932) [4]. However, now, it is not the experimenter or observer who chooses a measurement to make. The sequence of “measurements” being made is determined by the unitary evolution of the wave function of the universe; classical reality is realised step by step in a stochastic way with probability laws determined by the same underlying deterministic laws; the wave function undergoing repeated random partial collapses.

We see that the more simple situation in which *f* is not only onto but also one-to-one is so simple as to be completely uninteresting: the classical part follows a deterministic path, according to the iterates of the inverse of *f*; in each classical state a corresponding unitary evolution takes place of the state of the quantum part. The evolution can be termed autonomous in the sense that the classical world follows a deterministic path not influenced in any way by the state of the quantum world.

So the interesting situation is that *f* is onto but not one-to-one. This has some immediate consequences: first of all, it forces X to be infinite, and secondly, because $\sum_{x:f(x)=y}U_{xy}^{*}U_{xy}=1$, where the sum can be over several *x*, and at the same time $U_{xy}U_{xy}^{*}=1$ if f(x)=y, the matrix $U_{xy}$ is not itself unitary when *y* is the image of several *x*. Thus, the space L must be infinite dimensional.

Though the forward evolution of the classical part is stochastic, its backward history is deterministic: if *U* has been applied repeatedly bringing us into the classical state *x*, the classical history is given by the iterates of *f* on *x*. In terms of observables, $U^{*}$ maps classical observables to classical observables in the (reversed time) Heisenberg picture. The classical observables commute with everything, and can all be assigned values simultaneously. In particular, the whole past trajectory of the classical system up to the present time is itself classical.

These features of the model are retained when we drop the special product structure of the Hilbert space H. One point of this analysis is to show by elementary and direct means that the features exhibited by various toy models are generic to the approach. In particular, we can always identify some kind of shift operator—acting on classical trajectories—which is the source of the quantum-classical interaction in the model. The future of the trajectory is hidden in the quantum future; the past of the trajectory is fixed in the classical present.

## 3. Pedagogical Intermission: A Few Short Proofs from the Theory of von Neumann Algebras

In the following, A, C are always von Neumann sub-algebras of B(H) and *U* is a unitary operator on H.

1.Suppose C is commutative; suppose $UCU^{*}\subseteq C$. Then $A=C^{\prime }$ satisfies C⊆A, $A^{\prime }=C$, and $U^{*}AU\subseteq A$.

**Proof.** The commutant of a von Neumann algebra is also a von Neumann algebra; and its double commutant is itself. Since C is a commuting algebra it is obvious that its commutant must contain itself. This takes care of all assertions except the last. For that, we note that $UCU^{*}\subseteq C$ iff $C\subseteq U^{*}CU$ iff $A=C^{\prime }⊇{\left(U^{*}CU\right)}^{\prime }=U^{*}C^{\prime }U=U^{*}AU$. □

2.Suppose $A^{\prime }$ is commutative; suppose $U^{*}AU\subseteq A$. Then $C=A^{\prime }$ satisfies C⊆A, $C^{\prime }=A$, $UCU^{*}\subseteq C$.

**Proof.** The first assertions are again trivial; the last has been taken care of in the preceding proof. □

3.Let C be the set of operators of the form $\sum c_{x}\left|x\rangle \langle x\right|$ where the numbers $c_{x}$ are bounded and |x〉 is a countable orthnormal basis of H. Then C is a von Neumann algebra.

**Proof.** It is sufficient to check that $C^{\prime }=C$. So let *A* be a bounded operator which commutes with every element of C. The matrix elements 〈y|Az〉 determine *A*. We are given that for all *x*, |x〉〈x|A=A|x〉〈x|, hence for all *y* and *z* we have 〈y|x〉〈x|A|z〉=〈y|A|x〉〈x|z〉. Hence $\delta_{y=x}\langle x|A|z\rangle =\langle y|A|x\rangle \delta_{x=z}$. Taking y=x we see that 〈x|A|z〉=0 unless x=z. Because *A* is bounded, the numbers 〈x|A|x〉 are bounded, and A∈C. □

4.${\left(A⊗B\right)}^{\prime }=A^{\prime }⊗B^{\prime }$.

**Proof.** For A⊆B(H), and B⊆B(K), write A×B for the set of tensor products A⊗B, acting on H⊗K in the obvious way. We may now define A⊗B as the double commutant of A×B—the smallest von Neumann algebra containing all tensor products in A×B. Taking the commutant again, it follows that ${\left(A⊗B\right)}^{\prime }={\left(A\times B\right)}^{\prime \prime \prime }={\left(A\times B\right)}^{\prime }⊇A^{\prime }\times B^{\prime }$. Thus, ${\left(A⊗B\right)}^{\prime }⊇A^{\prime }⊗B^{\prime }$. The converse implication is rather more difficult to obtain—see Kadison and Ringrose, vol. 2. □

## 4. Some Examples

### 4.1. Representing a CP Map

One can embed an arbitrary CP map (taking quantum states to mixed classical-quantum) into eventum mechanics (allowing to extract both the measurement outcome and the transformed state). In this paper, we will only do a simple example, with *only* a shift on infinite chains of two-level systems.

The basic trick goes something like this. Let $H_{S}$ be the Hilbert space of the system being transformed and/or measured by the CP map. Consider an operator sum representation with matrices $A_{x}$ such that $\sum_{x}A_{x}^{*}A_{x}=1$; i.e., for simplicity x∈X and this outcome space is finite or countably infinite. The CP map produces the classical outcome *x* and transforms to the state $A_{x}\rho A_{x}^{*}$, normalized, with probability equal to the normalization constant, where ρ is the state (which is arbitrary) in $H_{S}$ of the system being transformed. I add to this a Hilbert space of the apparatus and a Hilbert space of the environment. The space of the apparatus will be simply $H_{A}=\ell^{2}\left(X\right)$. For the environment, take an infinite collection of copies of $H_{A}$, indexed by n∈Z. The tensor product of all these spaces is not separable but we restrict attention to a part of the space, namely $H_{E}$, the closure of the span of the countably many orthonormal vectors $|x_{n}:n\in Z\rangle$ for which all but finitely many of the coordinates $x_{n}$ are equal to a special value 0. To simplify the construction let us suppose that x=0 is not a possible value of the outcome of the measurement, i.e., $A_{0}=0$. Otherwise we simply extend X by adding one point different from those already present and call it 0. The environment component will be considered as the product of two parts, $H_{E}=H_{C}⊗H_{Q}$, by writing $|x_{n}{:n\in Z\rangle }_{E}=|x_{n}{:n<0\rangle }_{C}⊗{|x_{n}:n\geq 0\rangle }_{Q}$. We now have got a large, separable Hilbert space for system, apparatus and environment, where the environment again is the product of two parts thought of as classical and quantum, respectively. The algebra of observables of the joint system will be that generated by taking arbitrary quantum observables of the system, apparatus, and quantum environment, together with only classical observables (diagonal in the specified basis) of the classical environment.

The centre of this algebra can be identified with the classical observables on $H_{C}$. The classical states of this algebra correspond to infinite sequences of elements of X indexed by the negative integer $n\in Z_{<0}$, which only have a finite number of elements unequal to the special element 0.

The initial state of apparatus will be the state in which $x=x_{A}=0$, and that of the environment will have $x_{n}=0$ for all n∈Z.

We now describe a unitary mapping on the product system, as the composition of the following three maps, each working on different parts of the system. We describe the mapping in the Schrödinger picture, as unitary maps to be applied consecutively (on the left) to a vector in the large product system. First there is a unitary mapping of $H_{S}⊗H_{A}$ to itself satisfying ${|\psi \rangle }_{S}⊗{|0\rangle }_{A}\mapsto \sum_{x}|A_{x}{\psi \rangle }_{S}⊗{|x\rangle }_{A}$. As specified so far, the map preserves inner-products, and it can be extended in many ways to be unitary on the entire space $H_{S}⊗H_{A}$. This part will be very familiar as one of the many ways to express a CP map as a unitary mapping on a larger space followed by a projective measurement of part of the space (or by tracing out the complementary part).

Next we copy the measurement outcome, though still thought of as quantum (superposition is still possible) into the quantum part of the environment, and specifically the component n=0 of the quantum environment. The unitary achieving this can be taken to be any unitary taking ${|x,0\rangle }_{A,0}$ to ${|x,x\rangle }_{A,0}$ where the two indices stand for $H_{A}$ and the zero’th component of $H_{Q}$.

Finally we apply the unitary mapping to $H_{E}$ which performs the left-shift, taking $|x_{n}:n\in Z\rangle$ to $|x_{n+1}:n\in Z\rangle$. The composition of these three maps is called *U*. It operates as required on the observables of the joint system, since in the Heisenberg picture we first have the *right* shift, shifting the classical observable at position n=−1 into a larger quantum space, while the subsequent steps are unitary mappings acting on quantum observables only and leaving classical observables unchanged.

Back in the Schrödinger picture take the initial state of the combined system to be $\rho_{S}⊗{|0\rangle }_{A,E}$, where we abuse notation by writing just the state vector ${|0\rangle }_{A,E}$ rather than the corresponding density matrix. Applying *U* to this state converts it into the mixture, with probabilities $\mathrm{trace}\;\rho A_{x}^{*}A_{x}$, of the state which is the product of $A_{x}\rho A_{x}^{*}$, normalized, of the system, together with the pure state of apparatus and environment with $x_{A}=x$, $x_{E,n=-1}=x$, all other components equal to 0.

To make the description of the measurement make sense when *U* is iterated, we note that the apparatus contains a quantum memory of whether or not it has already been used, depending on whether $x_{A}=0$ or not. We can append to our prescription to steps 1 and 2 above, that when $x_{A}$ initially is not 0, nothing happens at all, while step 3 is unchanged. Alternatively, the classical part of the environment also contains a memory of whether or not a measurement already took place, so we can achieve the same effect by letting it control what happens in steps 1 and 2. In that case we could also delete one of our copies of $H_{A}$ and reduces steps 1 and 2 to a single step, by effectively taking the zero’th component to be the apparatus rather than part of the quantum environment.

After applying *U* any number of times, the classical environment is in a classical state which tells us when the measurement took place and what the outcome was; the probability of any particular outcome is what we require. After the measurement there is no entanglement of system, apparatus and environment; there is just a classical correlation. The system could be detached and measured another way in another measurement apparatus; the CP map we have implemented can also be seen as a quantum channel.

There are three unsatisfactory features of this model, apart from the huge amount of freedom which is left in how to completely specify the unitary maps involved.

The first is the huge size of the quantum environment—infinitely many copies of the apparatus space, even if the apparatus can be taken finite dimensional. However we know that we need to presume infinite dimensionality of the quantum part of the world, and the block-wise description which we have of possible *U* shows that something like a shift of an infinite sequence is going to be unavoidable. Since the model allows iteration of the unitary map, we have to allow infinitely many branches in the classical outcomes, and these have to be the reflection of an infinite possibility of branching in the quantum part. Moreover, every attractive measurement model so far discussed in the literature either needs an infinite system to begin with, or speaks about limiting properties of systems of larger and larger numbers of copies of finite systems. Such models typically give the nice results they do only in the large time limit. The Belavkin model could be thought of as an attempt to complete or extend the existing classes of models, so that behaviour which formerly could only be approximated arbitrarily well, can now also be exhibited exactly, inside the model.

The second unsatisfactory feature of the model is the fact that the initial state of the environment must be taken as fixed. Many attractive attempts to model measurement actually work by assuming an environment consisting of many small systems in a mixed state. One keeps inside a traditional framework with a unitary mapping on a completely quantum space of observables, generating classical probability at a macroscopic level from classical probability inserted at a microscopic level. These arguments usually involve some kind of averaging over microscopic degrees of freedom to destroy quantum coherence between different macroscopic states. Though physically appealing (Landsman quotes van Kampen as having said that someone who does not accept this does not understand what physics is) this argument is metaphysically very unsatisfactory, and still leaves the question open as to whether mathematically attractive models can be built within which the limit has been attained. However, at least the approach does respect the fact that a macroscopic apparatus and its even larger environment is never going to be in a very special, controlled, initial state. The answer to this from the Belavkin side must be that there do exist special states within macroscopic quantum systems. Is this the vacuum state of some quantum field? From this point of view, the quantum environment is actually at a very deep level inside of the systems being studied, and represents simply the effects of pure quantum noise from deeply microscopic levels.

The third apparently unsatisfactory feature from Landsman’s point of view is that the unitary mapping *U* is strongly related to (or constrained by) the chosen algebra of beables. Landsman would rather find the algebra of beables emerge from the description of *U*. That is indeed a feature of the toy models we have discussed, but in an abstract approach one is completely free to start from specification of *U* and then identify possible, and indeed incompatible, algebras of beables. For the same unitary mapping *U* there do exist different algebras with different centres, corresponding to rather different kinds of observers who are completely incompatible with one another, as Landsman would appreciate. We return to this discussion in Section 5.

### 4.2. The Geiger Counter

Basic books on quantum physics never give a model for the Geiger counter, supposed to give a click on the detection of a radioactive emission coming from a single atom. Yet this is presumably the apparatus in the Schrödinger cat story, which is supposed to detect whether or not an emission occurs in a certain time interval, and according to this trigger the poisoning or not of the cat. It is possible to give a simple Belavkin-type model for this situation, where an atom starts in the initial state α|0〉+β|1〉. With probability ${\left|\beta \right|}^{2}$ it delivers a macroscopic signal at a geometrically distributed random time (exponentially distributed in the continuous time limit), with probability ${\left|\alpha \right|}^{2}$ it never emits a signal. If after any time the atom has already given a signal, the atom will be in the state |0〉. If after a long time it still has not emitted a signal, it will still be in a superposition of |0〉 and |1〉, but with more and more weight on |0〉 as the length of time we have waited (and still nothing has happened) increases. The model has been worked out in detail by Feenstra (2009) [5], and by Brown (2021) [6].

### 4.3. Continuous Time

A decent (CP) continuous time measurement (and state transformation) process can also be represented in the Belavkin picture. The time shift operation becomes more natural than ever, the classical and quantum environment become larger but also more physically interpretable. It seems that as we scale up the model towards reasonable levels of space-time complexity, what initially seems like an excess of hidden layers in model, some of them too narrowly prescribed, others embarassingly free, comes into a decent balance with what has to be in the model anyway. Feenstra (2009) works out how the master equation from quantum optics can be neatly expressed as a Belavkin model, not surprisingly in view of Belavkin’s work on continuous time measurement and control of quantum systems in quantum optics.

## 5. Many Worlds?

Suppose we start with a Hilbert space H and a unitary *U*. We can now investigate whether or not there exist non-trivial von Neumann algebras C⊆A⊆B(H), where C is commutative, and such that $C^{\prime }=A$, $A^{\prime }=C$, $U^{*}AU\subseteq A$, and $UCU^{*}\subseteq C$. As we have mentioned before, half of these conditions follow from the other half, which leads to smaller sets of conditions whose starting point, which is a matter of taste, is either a commuting algebra C or a non-commuting algebra A.

Already, the toy models where H is a product, and *U* is built around a shift, show that different choices of C and A can indeed be compatible with the same H and unitary *U*: we can choose a different “preferred basis” of the first component of H. Changing the basis on the second component correspondingly, the shift remains a shift. Thus, we have a mathematical universe where Landsman’s many worlds and incompatible observables exist alongside Belavkin’s insistence on the non-demolition or causality principle for any specific class of compatible observers. It remains to be investigated whether “even more incompatible” worlds can exist than the ones which we can most easily locate in our toy models.

In our opinion Landsman does betray a physicist’s disregard of the reality of quantum jumps since apparently it is fine for them that one observer sees a universe evolving stochastically and irreversibly, while another sees completely different random jumps in an incompatible classical world. This seems to come down to a many worlds view where the many possible branches of the classical world all "exist" next to one another, in an ever increasing profusion, but are glued together and then teased apart into different incompatible branching strands by another incompatible observer. A realist of type B applies Occam’s razor, insisting that there is no use in considering different possible realities as equally real, if we only ever have access to one. Landsman’s (realist of type A) point of view is that we should not take an egocentric view of the observer. The best description of the world is not necessarily a picture largely coloured by our special position in it.

These questions remain metaphysical in eventum mechanics, since it accomodates realists of both persuasions. Whether it is type B or type A realists who are hallucinating cannot be determined.

We note that in approaches to the measurement problem to date, designed to show how a classical reality emerges from a quantum universe, already some notions of classicality are built in, in advance; for instance, a certain locality property is explicitly inserted in the Hepp model. As we mentioned before it is well possible that given H and *U*, partial information about C or A might be enough to derive other properties.

## 6. Discussion

A common picture of quantum measurement is that a quantum system under investigation comes into interaction with a large quantum system representing a measurement apparatus. At the end of the interaction, the apparatus is in a definite macroscopic state corresponding to what is usually called the position of a macroscopic pointer-variable. As Landsman made clear, the most convincing attempts to fill in the details in a way which is both physically meaningful and mathematically convincing require that one also considers what is called the environment, though he finished neither with a completed mathematical framework nor with complete examples. Rather, their analysis points to a collection of properties which would be desirable in a final model (whether general or specific). It is left open as to whether or not these properties are compatible and whether existing partial analyses can be completed on these lines. We have shown that Belavkin’s eventum mechanics satisfies Landsman’s requirements by assuming that the algebra of beables C is compatible in a precise sense with the unitary evolution of the universe *U*. Moreover, Belavkin’s framework describes exactly the same universe of possible quantum measurement processes as are usually considered in quantum information theory and which, in continuous time, turn up in diverse contexts in quantum measurement theory as seen by physicists, from the most applied and phenomenological to the most abstract, including earlier attempts to resolve the measurement problem by the addition of stochastic terms to the Schrödinger equation. Thus, from a mathematical point of view, there is no loss of generality in supposing that quantum measurement is described by eventum mechanics. What remains to be seen is whether the most interesting though so far partial attempts to model measurement through interaction with an incompletely knowable environment can be completed in an attractive way by expressing them in eventum mechanics.

The idea of adding an environment to a system-apparatus model, is that observers, who are also physical systems, cannot access all aspects of the environment. An observer by definition experiences a consistent (possibly stochastic) evolution of their (or her) restricted world. Landsman’s point of view is that we define an observer by defining an algebra of observables on our Hilbert space H. This algebra should not consist of *all* bounded operators but only a subset. The observables in the centre of the algebra, those which commute with everything in the algebra, define what the observer actually observes. This is essentially the abstract Belavkin picture with one exception: Landsman does not assume that the unitary evolution of the whole universe maps the observer’s observables into themselves. As a consequence, the classical worlds as seen by the observer at different time points are not consistent with one another. The observer does not have a memory; the classical past is not part of the classical present. Belavkin’s model allows the classical worlds at different times to mesh together properly exactly by their nondemolition or causality assumption. Landsman would appreciate the (mathematical) existence of incompatible observers who see the same physical laws of the same universe, i.e., the same Hilbert space and the same unitary evolution, but who live among different restricted algebras of observables.

Many physicists see the measurement problem as the problem of showing how a classical world, and preferably one particular one, emerges from a purely quantum description of the universe. In eventum mechanics, we start with a given time evolution (the unitary *U*). Many different classical worlds can be found which are “causally consistent” with this time evolution. It seems to us that commutativity should be looked for in quantum field theory where we introduce space as well as time, and express “locality” in the theory by the commutativity of different regions of space at the same time. Landsman also believes that locality is crucial to any solution of the measurement problem. We believe this is justified since it could single out those particular choices of C which are compatible with prior notions of time and locality. We believe it is necessary to adopt some peculiarities of the universe as seen from our point of view, in order to understand the emergence of the rest.

## 7. Conclusions

Klaas Landsman recently won the most prestigious Netherlands national science prize and accompanying research grant. The media announced that he was going to solve the measurement problem. I’m not up to date with the progress he already made since 1995, but I do believe he has a very good chance of getting that done before the prize money is used up and I wish them all possible good luck in that endeavour.

The hope of cosmologists is to develop a theory in which time and space itself should also “emerge” from a quantum theory of the universe. It seems to me that one should first have a firm understanding of how quantum theory does allow a classical world at all, with pre-existing notions of time and space, before embarking on this project. A similar remark can be made about the issues of reconciling relativity and quantum theory. In my opinion, theoretical physicists need to take causality seriously, and to take irreducible or intrinsic randomness seriously. These concepts need to be put into the ground level of physics. For Einstein, space and time were exchangeable, and the universe operated according to deterministic laws. For many physicists, the arrow of time is an emergent phenomenon. Could it be that these ideas are wrong?

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
