# Peer review of "Schrödinger’s Cat Meets Occam’s Razor"

_entropy, 2022, doi:10.3390/e24111586_

Round 1

Reviewer 1 Report

This is a very interesting perspective on the measurement problem in which the (old) conceptual framework of Landsman (1995) is combined with technical arguments by Belavkin (2007). It is much clearer than the work of these authors. In particular, the mathematical framework is explained very well, as is its interaction with a certain ideology. However, this combination does not, in my view, solve the measurement or Schrödinger Cat problem. Apart from various idealizations the author himself mentions and comments on in some detail, which may or may not be resolved by more advanced models, the main problem lies in  the interpretation of the equation just above line 183, i.e. the Schrödinger evolution of states. Of course, the mathematics as such gives this equation without the denominator trace (...) and the numerator trace (...), which of course cancel out and were added by the author in order to make an interpretational move that seems unwarranted to me by both the physics and the mathematics: "Thus the classical coordinate y jumps to one of the coordinates x ...". This admittedly agrees with the conceptual framework in Landsman (1995), who claims that states with a _unique_ decomposition in pure states admit a collapse interpretation with ensuing ignorance interpretation of the probabilities acting as coefficients (these are the trace ...). In that sense the author achieves his goals of matching the work of Landsman and Belavkin. But it is an arbitrary interpretational move, which Landsman (2017) has meanwhile disavowed in favour of a unitary collapse mechanisms described in detail in Chapter 11 of this book.   So I urge the author to add extensive comments explaining and justifying this move (which now goes entirely uncommented!), perhaps even in a more general framework than quantum mechanics. For example, he seems to regard it as a spontaneous collapse, in which case some reference to the traditional literature like von Neumann (1932) and its aftermath would be useful, and mention the fact that Landsman himself has meanwhile moved on. References: J. von Neumann (1932), Mathematische Grundlagen der Quantenmechanik (Springer);  K. Landsman (2017), Foundations of Quantum Theory (Springer, Open Access): http://www.springer.com/gp/book/9783319517766

Author Response

Reviewer 1

Response by RDG

I am very grateful for a very insightful review, and also for some kind words of praise: "Interesting perspective combining ..." and "much clearer than the work of these authors. In particular, the mathematical framework is explained very well, as is its interaction with a certain ideology".

Indeed, my aim in this paper was indeed to combine the work of Landsman (1995) with that of Belavkin (2007). And I do adopt a certain ideology as to what the measurement problem is, and what constitutes a solution. Incidentally, the Belavkin 2007 paper was a review of earlier work of that author. Belavkin's "solution of the measurement problem" actually predates the Landsman (1995) paper. Here are two earlier Belavkin papers where the same ideas and same mathematics can be found, especially the 1994 paper in Foundations of Physics with the introduction of his "nondemolition principle":

Nondemolition Principle of Quantum Measurement Theory. Foundations of Physics, 1994, 24, No. 5, 685–714.

Quantum Causality, Decoherence, Trajectories and Information, Rep. on Prog. on Phys, 2002, 65:353–420

I agree with the referee that I should have stated clearly that Landsman himself has moved on from the ideas developed in his 1995 paper, more than 25 years old by now. I have made this clear in the revision of the paper and added reference to his 2017 book, but I do not want to discuss his 2017 treatment of the measurement problem. It is an excellent discussion, but I wish to advocate a different "ideology". Landsman has recently received a major science prize in the  Netherlands centered on his project to "solve the measurement problem" in the next five years. Clearly, the last word on the measurement problem has not been said.

I now turn to the "hard core" of the reviewer's comments. He says "this combination, does not, in my view, solve the measurement or Schrödinger Cat problem".

I would respond that what he says is a legitimate point of view. But it is "merely" a point of view. It is an opinion. Well-founded, to be sure, but not the unique possible opinion. In my view, the combination does solve the problem. It all depends on what you think is the problem (or even: whether or not you think there is a problem), and what you desire of a "solution".

According to the reviewer, the approach does depend on what he calls an "arbitrary interpretational move". I agree that it is an interpretational move, but that is the name of the game: a solution of the measurement problem must depend on some interpretational move. I disagree strongly that it is an *arbitrary* move.

To start with, coming myself from probability and statistics, I find a probability interpretation extremely natural and appealing. I even find it *necessary*.

Quantum mechanics predicts probabilities via the Born rule, and it seems to me that we need the Born rule when we investigate whether experimental results follow  theoretical predictions, and we appeal to it when we build quantum informational technology and sell it to users. 

The referee writes "the mathematics as such gives this equation without the denominator trace (...) and the numerator trace (...), which of course cancel out and were added by the author in order to make an interpretational move that seems unwarranted to me by both the physics and the mathematics".

I disagree that this move is "unwarranted" by the mathematics. It is *allowed* by the mathematics, and it then allows a physical interpretation, namely of a probabilistic jump. I find it a necessary move. The aim of this work is to build a model of a small self-contained quantum universe which is ruled by quantum mechanics and which lives a stochastic life. Note that cosmologists who simulate possible actual universes do actually use repeatedly the Born rule in order that a random mass distribution of galaxies and stars arises from an early universe, and such that some realisations produce universes at a later timepoint looking much like ours, some living much shorter, some much longer.

Many efforts have been made to somehow derive the Born rule from the "deterministic" part of quantum mechanics. In my opinion, none of them are compelling. One may certainly argue that the Belavkin approach is merely an attempt to revive some of the ideas of the founding fathers of quantum mechanics. The "Heisenberg cut" was named after Heisenberg and corresponds to separating a physical situation into a classical and a quantum part. Bohr argued again and again that one needed to use conventional classical physical language to describe the experimental set-ups in which phenomena could be observed (described in classical terms) caused by a quantum behaviour below the surface, where totally different physical laws held. Belavkin's approach shows that the Heisenberg cut can be incorporated into the framework not as an artificial add-on but as part of the fundamental quantum stochastic dynamics of the classical world which we see around us.

I have added a new paragraph explaining the relation between the key formula of the paper and traditional Von Neumann collapse. The formula is now numbered (1). The right hand side is a joint state of a composite system with a classical and a quantum component. Such states can be decomposed in a unique way into marginal and conditional states on the two components. The "move" is not just one of many optional interpretational moves. It is uniquely determined by what we are talking about - a toy universe with a classical past and a quantum future, evolving stochastically in time.

Reviewer 2 Report

The manuscript is not a research paper, and not even an historical one: as such, it is not appropriate for the journal. More than a paper, the manuscript is a “chat” about some works of Slava Belavkin, presenting no new result, but mentioning some known mathematical identities and/or theorems related to the work of Belavkin, however, without providing novel/shorter rigorous derivation. 

Author Response

Reviewer 2

Rev2: "The manuscript is not a research paper, and not even an historical one: as such,  it is not appropriate for the journal."

RDG: That is a matter of opinion. I understand that many physicists might have the same opinion as Reviewer 2. However,  Reviewers 1 and 3 clearly have quite different opinions.

My paper is deliberately aimed at scholars and scientists working in the foundations of physics / philosophy of science. In these fields, substantial academic debates (in the good sense of the word "academic") occur in which the mathematical level must be kept as low as possible while still allowing substantial discussion of the philosophical / metaphysical meaning of mathematical-physical assumptions.

Rev2: "More than a paper, the manuscript is a “chat” about some works of Slava Belavkin,  presenting no new result..."

RDG: My paper does not aim to present new mathematical results. It is intended to make Belavkin's contributions (the conceptual ones, not the technical mathematical ones) widely accessible to a much wider range of interested scientists and scholars. Nobody ever did that before! Belavkin's ideas are disappearing into obscurity. That fact is partly due to Belavkin's idiosyncratic notations and intensive use of rather heavy mathematical apparatus, not generally well known, or not recognised by non-specialists. Maybe Reviewer 2 thinks that Belavkin's solution to the measurement problem is not interesting? Some people will certainly think so.

Rev2: "...but mentioning some known mathematical identities and/or theorems related to the work of Belavkin, however, without providing novel/shorter rigorous derivation."

RDG: The reviewer has apparently not bothered to look for any actual mathematical errors in my paper. I claim that my mathematics is as rigorous as necessary in order to  faithfully reproduce what Belavkin's approach says, in the case of discrete time and separable Hilbert space (countable basis). This was not easy work at all. Once done, the maths is simple. That was my aim. I don't think it was obvious in advance that it could turn out to be so simple. I do believe that my presentation preserves the most important conceptual content of Belavkin's solution to the measurement problem in the least technical possible context.

RDG: Notice that Reviewer 1 and Reviewer 3 disagree entirely with Reviewer 2's  *opinions*. Obviously, the paper could be a "boundary case" for the journal "Entropy". It is certainly not the usual kind of paper which Entropy publishes. But I do not  believe that it falls outside of the editorial policies of the journal. I gladly submit to the scientific editor's judgement on this.

Clearly, the editors will have to show some daring and personal taste to accept my paper. I humbly suggest that "Entropy" is the very best choice of journal in which to publish such work. Belavkin's approach to the measurement problem is to see it as a mathematical problem which has a mathematical solution. Actually, there are many solutions, since the abstract mathematical framework does not tell the physicist where to place the Heisenberg cut. But any mathematical framework comes with the same issue: how to map mathematical objects to concepts of things in the physical world. One should see this as a feature, rather than as a bug.

Reviewer 3 Report

The work is very unusual. It is difficult to indicate the scientific result in it, but it probably was not the author's goal.
In the article, the author tries to introduce mainly the theory of measurement from the point of view of Landsman and Belavkin’sworks.

The work is full of many interesting insights and comments and indeed makes the above-mentioned results more understandable to non-specialists.

For this reason, I believe that the work can be published. Personally, I think that there is too little such work. On the other hand, I have doubts as to whether this article is suitable for Entropy, but that should be decided by the editors.

Author Response

Reviewer 1: 

The work is very unusual. It is difficult to indicate the scientific result in it, but it probably was not the author's goal. In the article, the author tries to introduce mainly the theory of measurement from the  point of view of Landsman and Belavkin’s works. The work is full of many interesting insights and comments and indeed makes the above-mentioned results more understandable to non-specialists.

For this reason, I believe that the work can be published. Personally, I think that there is too little such work. On the other hand, I have doubts as to whether this article is suitable for Entropy, but that should be decided by the editors.

RDG:

It should go without saying that I was delighted with this report.

I do present the point of view of the theory of measurement especially as elaborated in several major papers, which are nowadays little known, by Belavkin. I show their  relation to work of Landsman. Landsman himself has now "moved on" but I believe that Belavkin's approach is still intellectually and mathematically viable and not in conflict with anything we presently know for sure in physics. 

Obviously, what I just said are personal opinions, and as such, others may have different opinions. However, it seems to me to be intellectually very important to keep one's options open. As a mathematician and a scientist I do not have to choose one interpretation of quantum mechanics. I try to understand the consequences of all  interpretations and the interrelations between them. I do find Belavkin's approach rather attractive and presently not well known; I think that is due to Belavkin's personal approach and to the fact of his much too early passing.

Round 2

Reviewer 1 Report

None